# Biochar Facilitated Direct Interspecies Electron Transfer in Anaerobic Digestion to Alleviate Antibiotics Inhibition and Enhance Methanogenesis: A Review

**DOI:** 10.3390/ijerph20032296

**Published:** 2023-01-27

**Authors:** Kaoming Zhang, Yuepeng Deng, Zhiquan Liu, Yiping Feng, Chun Hu, Zhu Wang

**Affiliations:** 1Key Laboratory for Water Quality and Conservation of the Pearl River Delta, Ministry of Education, Institute of Environmental Research at Greater Bay, Guangzhou University, Guangzhou 510006, China; 2Guangdong Key Laboratory of Environmental Catalysis and Health Risk Control, Institute of Environmental Health and Pollution Control, School of Environmental Science and Engineering, Guangdong University of Technology, Guangzhou 510006, China

**Keywords:** anaerobic digestion, biochar, DIET, mechanisms, antibiotics

## Abstract

Efficient conversion of organic waste into low-carbon biofuels such as methane through anaerobic digestion (AD) is a promising technology to alleviate energy shortages. However, issues such as inefficient methane production and poor system stability remain for AD technology. Biochar-facilitated direct interspecies electron transfer (DIET) has recently been recognized as an important strategy to improve AD performance. Nonetheless, the underlying mechanisms of biochar-facilitated DIET are still largely unknown. For this reason, this review evaluated the role of biochar-facilitated DIET mechanism in enhancing AD performance. First, the evolution of DIET was introduced. Then, applications of biochar-facilitated DIET for alleviating antibiotic inhibition and enhancing methanogenesis were summarized. Next, the electrochemical mechanism of biochar-facilitated DIET including electrical conductivity, redox-active characteristics, and electron transfer system activity was discussed. It can be concluded that biochar increased the abundance of potential DIET microorganisms, facilitated microbial aggregation, and regulated DIET-associated gene expression as a microbial mechanism. Finally, we also discussed the challenges of biochar in practical application. This review elucidated the role of DIET facilitated by biochar in the AD system, which would advance our understanding of the DIET mechanism underpinning the interaction of biochar and anaerobic microorganisms. However, direct evidence for the occurrence of biochar-facilitated DIET still requires further investigation.

## 1. Introduction

Anaerobic digestion (AD), a cost-effective technology to convert organic waste into clean renewable energy, has been globally deployed for organic waste treatment and biogas production [1,2]. The organic carbon from wastewater or solid waste, such as livestock wastewater and manure, can be efficiently converted into biogas using AD treatment, the composition of biogas is about 40–70% methane and 30–60% carbon dioxide and small amounts of gaseous water, hydrogen sulfide, and ammonia [3], which is a promising replacement for traditional fossil fuels because of high calorific value (21–25 MJ/m^3^) [4]. Generally, the biological process of AD is separated into four steps: hydrolysis, acidogenesis, acetogenesis, and methanogenesis [4]. Hydrolysis can be described as complex organic matter hydrolyzed into soluble monomers through the action of microbial extracellular hydrolytic enzymes. The soluble simple monomers from hydrolysis are metabolized to fatty acids, hydrogen, and carbon dioxide by the fermentative bacteria during acidogenesis. Acetogenesis is a symbiotic process between anaerobic bacteria and methanogens that assimilates the compounds produced from acidogenesis to generate acetate. In the last step, methanogenesis is a rate limitation step, because it is the slowest biochemical reaction, and also a crucial step as all the compounds from the previous stages are converted to methane [4,5]. Despite the unique advantages of AD for the bioconversion of organic waste to biogas, it has certain limitations. For example, biogas enriched with methane and carbon dioxide is considered a greenhouse gas (GHG), and uncontrolled emissions have caused serious environmental risks [6]. Microbial conversion of organic waste into value-added products such as biofuels, biofertilizers, and biodegradable polymers through AD is a promising strategy [5]. In addition, although AD is very beneficial for the treatment of various organic wastes, there is a vulnerable balance between anaerobic microbial communities in the four steps. For instance, the acidogenesis step generates a mass of hydrogen that may lead to the increase of hydrogen partial pressure higher than 10 Pa, which would destroy the balance between acidogenesis and methanogenesis resulting in the failure of the AD process [7]. Furthermore, the interspecies hydrogen/formate transfer between anaerobic bacteria and methanogens is vulnerable to a variety of environmental factors such as volatile fatty acids (VFAs) and ammonia [8]. These drawbacks potentially result in instability, inefficiency, and even failure of the AD process [9,10]. Moreover, microbial inhibition by antibiotics has been suggested as the other main reason that decreases the efficiency of the AD system [11,12]. Antibiotics can weaken the ability of microbes to transform complex organic matter into soluble monomers by disrupting cell wall structure, inhibiting protein synthesis, and interfering with cell membrane function [13]. Aydin et al. showed that methanogens are extremely sensitive to the accumulation of antibiotics, resulting in an accumulation of VFAs and a decrease in methane production [14]. Antibiotics are widely designed and employed in the fields of medicine, breeding, and livestock to inactivate or suppress the growth of microorganism [15,16]. After administration, only 10–50% of antibiotics are metabolized or absorbed, and most of the original drugs and their metabolites are excreted via urine and feces [17,18]. Therefore, antibiotic residues from the production of agriculture and livestock accumulate within the feedstock, which is harmful to the AD system [19].

Recently, direct interspecies electron transfer (DIET), which is based on the syntrophic metabolism of microbial species that can be implemented without relying on electron carriers such as hydrogen or formate, has been recognized as an effective strategy for improving the stability and alleviating antibiotic inhibition in AD systems [2,20]. Studies have indicated that DIET exists in a wide range of microbial communities and is of global significance in the biogeochemical cycle [21]. It is considered that DIET contributes to improving the production of methane in anaerobic digesters [22,23]. In addition, DIET can facilitate the bioremediation of organic contaminants in soils and waters [24]. Meanwhile, it was revealed that an efficient DIET process in AD not only decreased the inhibition of antibiotics but accelerated the syntrophic metabolism of antibiotics [25,26]. Wang et al. have demonstrated that the potential DIET process contributed to enhancing the biodegradation of antibiotics and decreasing the intensive excretion of extracellular polymeric substances (EPS) under antibiotic pressures [25]. At present, three hypotheses have been proposed for the electron transfer pathway in DIET consortia (Figure 1), i.e., electron transfer between microbial species via outer-membrane c-type cytochromes, conductive pili, and/or conductive materials [27,28]. Among them, the outer-membrane multiheme cytochromes are redox substances, mediated by consecutive c-type cytochromes, and the conductive pili are mainly facilitated by metal-like conductive structure [27,29,30]. Furthermore, conductive materials, such as biochar and magnetite, also shuttle the electron between microbial species through the DIET mechanism [10,31]. Biochar, a convenient and cost-effective conductive carbon-rich material produced by the pyrolysis of biomass in an oxygen-limited atmosphere [32], has been widely used in enhancing AD performance [33,34,35]. Cui et al. have indicated that biochar surface enriched *Methanosarcina* and *Syntrophomonas* species were probably involved in DIET with biochar acting as an electrical conduit [36]. Similarly, it was shown that biochar could act as an electron shuttle to mediate the DIET process of *Methanosarcina* and *Geobacter* [31]. Nevertheless, the potential DIET mechanism for facilitating methanogenesis and contaminant removal is still unclear. Although most experimental evidence has indicated the contribution of DIET, the direct observation of electron flow between syntrophic microbes is still lacking [37]. Hence, it is important to discuss the underlying mechanisms of biochar-facilitated DIET for AD enhancement.

In this paper, we introduced the evolution of DIET and sorted out the application of biochar to facilitate DIET for alleviating antibiotic inhibition and enhancing methanogenesis over the years. Subsequently, the potential mechanisms of biochar-facilitated DIET for the improvement of AD performance in terms of electrochemical characteristics of biochar on DIET stimulation, as well as the effect of biochar on microorganisms, were illustrated. Ultimately, the prospects of biochar-facilitated DIET for AD technology were discussed.

## 2. The Evolution of the DIET

DIET was first discovered within the aggregate of an evolved syntrophic coculture of two *Geobacter* species (*Geobacter metallireducens* and *Geobacter sulfreducens*) [20]. *G. metallireducens* could use ethanol as an electron donor but was unable to reduce fumarate; conversely, *G. sulfurreducens* could not use ethanol but was able to reduce fumarate [20]. Tight spherical aggregates were formed between *Geobacter* species in the culture with ethanol as the electron donor and fumarate as the electron acceptor [20]. When *G. sulfurreducens* was replaced with a wild type that lacked gene *hyb* encoding hydrogenase, the growth of aggregates did not slow down but increased [20]. However, by deleting genes that encode *omcS* and *pilA*, which are related to c-type cytochrome and pili from them, the growth of aggregates slowed down dramatically [20]. Hence, Summers et al. discovered that there exists an extracellular electron transfer mode via pili and c-type cytochrome [20]. Conductive pili are also known as microbial nanowires [38]. Through microbial nanowires, electrons generated by the intracellular metabolism of microorganisms could be transported over long distances to the extracellular receptors, altering the recognition of electron transfer as being restricted to the intracellular compartment [38,39]. Conductive pili-mediated DIET had been broadly reported in wastewater treatment, bioremediation, and methanogenesis [40,41,42]. Li et al. revealed that *Pseudomonas* and *Sporanaerobacter*, which were enriched in *pilA*, were capable of DIET-based syntrophic metabolism by the utilization of complex organic compounds as electron donors directly [40]. Yin et al. reported a remarkably high abundance of *pilA* in the syntrophic coculture of *Geobacter*, *Methanosaeta*, and *Methanosarcina*, implying the potential involvement of conductive pili-mediated DIET in methane production [41].

To date, the role of DIET-associated genes has been further investigated. Wang et al. reported that the function of *pilA* was to modulate the secretion of *omcS*-encoded nanowires of protein together with other multiheme cytochromes and over-expression of *pilA* was accompanied by the excessive production of *omcS*-encoded protein nanowires [43]. Microorganisms are equipped with various redox proteins for mediating electron transfer, but the types of redox proteins contributing to various electron transfer models are distinct [44]. Electron transfer might occur through outer-membrane c-type cytochrome and other redox proteins such as multi-copper proteins (*OmpB* and *OmpC*), pilus-like nanowire structures, and extracellular or self-excreted small molecule electron shuttles (e.g., phenazines, flavins, and quinones) [45]. It was shown that *G. sulfurreducens* possessed 111 genes coding for c-type cytochrome [45]. *G. sulfurreducens* outer-membrane c-type cytochrome-mediated DIET was affected by a variety of genes, including *omcB*, *omcE*, *omcS,* and *omcZ* [45]. The multiheme c-type cytochrome played a critical role in reducing extracellular metal oxides by *G. sulfurreducens*, and the deletion of either *omcS* or *omcE* genes would decrease the reductive capacity for manganese oxide and ferric iron [46].

In addition, several conductive materials, such as magnetite, graphene, and biochar, were also found to be ideal additives to facilitate DIET in syntrophic metabolism [10,31,47]. Lin et al. indicated that graphene significantly improved the performance of AD by facilitating DIET [47]. Zhang et al. showed that the improvement in AD performance was primarily attributed to the enhancement of the DIET process by nano-magnetite to facilitate the syntrophic metabolism of organic matter [48]. However, carbon-based nanomaterials are usually prone to wash away and are difficult to recycle, while metal-based nanomaterials possibly result in material agglomeration, leading to a significant reduction in utilization efficiency [49,50,51]. Simultaneously, both the toxicity effect of nanoparticles on microorganisms and the high cost of nanomaterials hinder its application [52]. Wang et al. summarized that biochar is more effective at enhancing AD performance than carbon nanotubes and graphene [37]. Therefore, micron to millimeter scale biochar has been considered as the better additive for enhancing AD in practical applications due to its excellent performance, low toxicity and cost, as well as easy-recycling.

## 3. Application of Biochar-Facilitated DIET for Alleviating Antibiotics Inhibition and Enhancing Methanogenesis

Extensive residues of antibiotics in livestock manure have become a hot topic of current research [17]. In China, 52% of total antibiotics were used for veterinary purposes [15]. As a promising technology for livestock manure treatment, the removal of residual antibiotics from manure with AD has been widely explored [25]. Unfortunately, it had been shown that antibiotics reduce the stability and efficiency of the AD system greatly [18]. Recently, many studies have confirmed that an enhanced DIET process with biochar can relieve the pressure of antibiotics in the AD system [53,54]. DIET is an excellent pathway for improving the efficiency of electron transfer involved in the syntrophic metabolism of microorganisms. Numerous studies have shown that the DIET process could enhance methane production and shorten lag time [2,31].

In Table 1, we summarized some studies on biochar-facilitated DIET for alleviating antibiotics inhibition and enhancing methanogenesis. Wang et al. found that biochar speeded syntrophic methanogenesis via facilitating the emergence of the DIET process at high TC (50 mg/L) pressures; meanwhile, the addition of biochar improved the TC removal efficiency by 34.2% [25]. The improvement of antibiotic removal efficiency was primarily attributed to the enhancement of the microorganisms with contaminant degradation capacity, rather than the physiochemical adsorption of biochar [25]. Cheng et al. indicated that the removal of sulfadiazine (SDZ) and sulfamethazine (SMZ) increased by 25.77% and 26.46%, respectively, during the 13 days after the addition of 0.5 g/L of biochar [55]. However, there was no significant variation in methane production, probably owing to the low dose of biochar addition (only 0.5 g/L) [55]. Previous reports suggested that the enhancement capacity of biochar on methanogenesis seemed to be positively correlated with the dose of biochar [56]. Zhang et al. revealed that at doses of 6.2, 15.9, and 26.1 g/L of biochar, the cumulative methane yield from the three reactors was raised by 17.80%, 46.99%, and 57.47%, respectively [56]. Yang et al. showed that the addition of biochar at an optimum dose of 5–10% (based on total solids of manure) dramatically boosted the yield of methane by 23.6–25.1%, which was attributed to accelerating the conversion of VFAs to methane by the biochar-facilitated DIET processes [1].

The existence of microbes involved with DIET could improve the stability and efficiency of the AD system [37,42]. A study by Li et al. indicated that biochar with homogeneous dispersion contributed to a more robust DIET process between microbes, in which aromatic functional groups on the biochar surface were potentially directly involved in the DIET process [57]. The rich functional groups and high aromatic content of biochar contributed to bacteria accepting electrons from syntrophic microbes [58,59]. Bu et al. showed that biochar with quinone groups had a shortened lag time and significantly increased methane production by 45.9% [35]. Lei et al. found that redox-active biochar of 4.0 g/L significantly increased the biotransformation of tetracycline (TC), triclocarban (TCC), triclosan (TCS), and sulfamethoxazole (SMX), resulting in a remarkable improvement of 13.7% in the removal efficiency of organic micropollutants [60]. The adsorption performance of carbon-based materials was generally regarded as the primary contributor to enhancing the removal efficiency of organic micropollutants [61]. However, Wang et al. demonstrated that the reinforcement of the electron transfer system of microorganisms by quinone and hydroquinone groups of biochar was probably the mechanism of enhanced biotransformation of organic micropollutants [62]. Wang et al. found that biochar effectively alleviated acidification in the reactor after a high organic loading shock (OLS) of 80 kg COD/m^3^/d [31]. After a period of 24 h at a high OLS, the reactor without biochar experienced irreversible acidification (pH = 5.42) and biogas production was merely 0.08 m^3^/kg COD/d. In the case of the reactor with 4 g/L of biochar, biogas production recovered rapidly to 0.33 m^3^/kg COD/d, while the pH increased to 7.01 [31]. Biochar remarkably enriched potential DIET microbes such as VFAs-oxidized bacteria (*Bacteroidetes*, *Smithella*, *Desulfovibrio*, *Geobacter*) and methanogens (*Methanosaeta*, *Methanosarcina*), maintaining the balance of acidogenesis and methanogenesis in the reactor [31]. Li et al. showed that 10 g/L of biochar significantly improved methane production by 44%, with methane production efficiency up to 0.24 m^3^/kg COD [2]. Biochar stimulated the growth of some DIET microbes, such as *Geobacter*, *Sphaerochaeta,* and *Sporanaerobacter* species, which further boosted the DIET-based syntrophic metabolism to degrade organic compounds and generate more methane [2]. Zhang et al. showed that 5 g/L of biochar greatly alleviated sulfamethazine (SMZ) inhibition through biochar adsorption in the early period of the AD process (0–3 days), and only 0.6% of cumulative biogas production decreased at 20 mg/kg^−1^ of SMZ (based on TS of manure) [26]. At the same time, the microbial community analysis indicated that biochar significantly facilitated the enrichment of microbes involved in DIET such as *Methanothrix* and *Methanosarcina* [26].

**Table 1 ijerph-20-02296-t001:** Application of biochar-facilitated DIET for alleviating antibiotics inhibition and enhancing methanogenesis.

Biochar Feedstock	Biochar Preparation/Dosage	Biochar Properties (Size, BET)	AD Feedstock	Experimental Scale	Operating Condition	Antibiotics (Concentration)	Removal Rate	AD Performance	Reference
Sawdust	500 °C, 1.5 h/ 15 g/L	0.25–1 mm, 38.6 m^2^/g	Swine manure	Laboratory-scale	35 °C 120 rpm	Tetracycline (0.5, 50 mg/L)	79.7% 55%	CH_4_ production rate inhibited by 12.5% under 50 mg/L TC pressure	[25]
Rice husk	300 °C, 2 h/ 5–10% of dry manure	0.25 mm, 16.66 m^2^/g	Swine manure	Laboratory-scale	37 °C 170 rpm	-	-	Methane yield increase 23.6–25.1%	[1]
Citrus peel	500 °C, 1 h/ 1.5 g/g vs. of substrate	0.25 mm, 6.6 m^2^/g	Sewage sludge Food waste	Laboratory-scale	35 °C 120 rpm	-	-	Facilitated methane production (250.8 mL/g VS) and shortened lag time (3.5 days)	[63]
Pomelo peels	600 °C, 2 h/ 0.5 g/L	0.075 mm, 27.50 m^2^/g	Synthetic swine wastewater	Laboratory-scale	35 °C HRT:22 h	Sulfadiazine Sulfamethazine (100 μg/L)	25.77% 26.46%	Remove 95% of COD with average methane yield of 0.2 L/g COD	[55]
Fallen leaves	300 °C, 2 h/ 15 g/L	0.15 mm, 3.8 m^2^/g	Sewage sludge	Laboratory-scale	37 °C 150 rpm	-	-	Shortened lag time and increased 45.9% methane production	[35]
Corn straw	600 °C, 0.3 h/ 4% of TS	0.5 mm, -	Corn straw	Pilot-scale	38 °C 60 rpm	-	-	Biogas yield increase 19.42%	[64]
Wheat straw	500 °C/ 10 g/L	10–15 mm, -	Kitchen wastes Waste activated sludge	Laboratory-scale	37 °C 30–40 rpm	-	-	Methane production increase 44% and methane production efficiency of 0.24 m^3^/kg COD	[2]
Cattle manure Sawdust	300 °C 500 °C 700 °C, 1.5 h/ 15 g/L	0.25–1 mm, 4.2–181.3 m^2^/g	Food wastes Sewage sludge	Laboratory-scale	35 °C 120 rpm	-	-	Lag time decreased from 15.0 d to 1.1–3.0 d, maximum CH_4_ production rate increased from 4.0 mL/d to 10.4–13.9 mL/d	[62]
Sawdust	500 °C, 1.5 h/ 15 g/L	0.25–1 mm, 248.6 m^2^/g	Food waste Activated sludge	Laboratory-scale	35 °C 30–40 rpm	-	-	Shortened lag time by 39.2–52.8%	[65]
Sawdust	-/0.60 g/g vs. oily sludge	-, 840.68 m^2^/g	Naphthalene Starch Oily sludge	Laboratory-scale	35 °C 110 rpm	-	-	Maximum CH_4_ yield (138.41 mL/g VS) was 2.19 times of control	[66]
Hickory wood chips	900 °C/ 12 g/L	-,-	3.2 g ethanol/L/d	Laboratory-scale	36 °C	-	-	Methane production improved by 75% and specific methane production increased to 725 mL/g VS/d	[58]
Corncob	500 °C, 1.5 h/ 4 g/L	2–3 mm, 37.8 m^2^/g	Synthetic sewage	Laboratory-scale	35 °C HRT:3.2 h	Tetracycline Triclocarban Triclosan Sulfamethoxazole (0.2 μg/L)	-	Improved the biotransformation potential of antibiotics, increasing the overall removal efficiency by 13.7%	[60]
Corncob	500 °C, 1.5 h/ 2 g/L	2–3 mm, 9.4 m^2^/g	Synthetic sewage	Laboratory-scale	8 °C 10 °C HRT:4 h	-	-	CH_4_ production increased by over 15% at 10 °C	[67]
Corn straws	900 °C, 1 h/ 10 g/L	0.18 mm, -	Glucose Food waste	Laboratory-scale	37 °C 80 rpm	-	-	Cumulative methane production improved by 42.07%	[68]
Apple tree branch	550 °C/ 100% of dry sludge	0.18 mm, 19.6 m^2^/g	Excess sludge	Laboratory-scale	37 °C	-	-	Cumulative methane production was 172.3 mL/g COD, 30.2 times of control group	[69]
Cotton- wood	700 °C, 0.6 h/ 8 g/L	0.5 mm, 13.97 m^2^/g	Cornstalk	Pilot-scale	45 °C 60 rpm	-	-	Improved the volumetric biogas production rate by 43.09%	[70]
Iron-rich sludge	700 °C, 1.5 h/ 10 g/L	0.048 mm, -	Synthetic wastewater	Laboratory-scale	35 °C 140 rpm	-	-	Cumulative methane production of 486 mL/L and maximum methane production rate of 2.0 mL/h	[71]
Wood chip	850 °C/ 8 g/L	-, 272 m^2^/g	Municipal Solid Waste	Pilot-scale	51–53 °C -	-	-	Improved the methane content by 10%	[72]
Corn stalks	500 °C, 2 h/ 5 g/L	0.55 mm, -	Manure Corn stalk	Laboratory-scale	37 °C	Sulfamethazine (60 and 120 mg/kg^−1^ of dry sludge)	97.6% 96.8%	Cumulative biogas yield decreased by 7.2% and 8.7%, respectively	[26]
Excess sludge	400 °C, 2 h/16% of dry sludge	-,-	Excess sludge	Laboratory-scale	35 °C	-	-	CH_4_ production increased 54.5%	[73]
Blue algae	450 °C, 2 h/ 10 g/L	0.1–0.3 mm, 145.6 m^2^/g	Sludge hydrolysate with 1% and 4% (*v*/*v*) inoculation ratio	Laboratory-scale	35 °C 120 rpm	-	-	Methane production increased by 12.2% and 17.5% and lag times shortened by 41.6% and 44.3%, respectively	[74]
Wood pellets	700–800 °C/ 15 g/L	0.2–0.3 mm, 193 m^2^/g	Food waste	Pilot-scale	55 °C	-	-	Improved the average methane production by 37%	[75]
Discarded fruitwood	550 °C, 2 h/ 3.3% of dry sludge	0.3–0.45 mm, 206 m^2^/g	Chicken manure	Laboratory-scale	35 °C	-	-	Maximum cumulative methane production of 294 mL/g vs. manure	[76]
Pine sawdust	900 °C, 0.33 h/15 g/L	0.025 mm, 265 m^2^/g	Food wastes	Laboratory-scale	37 °C	-	-	Increase 46.9% cumulative methane production and 43.0% daily methane production rate	[77]

## 4. Mechanisms of Biochar-Facilitated DIET for Alleviating Antibiotics Inhibition and Enhancing Methanogenesis

### 4.1. Electrochemical Mechanism of Biochar-Facilitated DIET

Electron transfer via the DIET process is a complex electrochemical process. The electrical conductivity, redox-active characteristics, and electron transfer system activity can be used to investigate the electrochemical mechanism of biochar-facilitated DIET.

#### 4.1.1. Electrical Conductivity

Electrical conductivity was commonly employed to characterize the electron transfer ability of specific materials, with higher conductivity suggesting that more electrons were transferred per unit of time [37]. Previous studies showed that the conductivity of methanogenic aggregates usually resulted from significantly enriched *Geobacter* species, which induced the formation of conductive nanowires and/or c-type cytochromes *OmcS* that make biomass conductivity [78,79]. It was suggested that the physical and electrical properties of carbon-based materials contributed to the enhancement of the electrical conductivity of microorganisms [37,80]. Enhanced electrical conductivity was commonly considered to facilitate the DIET process in syntrophic microbes. Studies showed that carbon-based materials improved the conductivity of the AD system for facilitating the electron transfer between syntrophic microbes, and thereby increased methane production [37]. The electrical conductivity of sludge could be a characteristic for evaluating the enhancement of electron transfer capability [81,82]. Zhu et al. showed that biochar prepared at the pyrolysis temperatures of 600 °C and 1000 °C improved the electrical conductivity of sludge by 95.3% (32.54 µS/cm) and 69.3% (28.21 µS/cm), respectively [83]. Wang et al. found that 4 g/L of biochar improved the sludge conductivity from 12.31 µS/cm to 23.29 µS/cm, with a higher sludge conductivity of 89.20% than the control group (11.62 µS/cm), and shortened the lag time by 28.6% [84]. Zhu et al. found that 10 g/L of biochar dramatically facilitated the biodegradation of organic matter by over 80%, while biochar increased the electrical conductivity of sludge by 1.9 times [85]. Biochar could be used as the growth carrier for microbes to further induce the formation of conductive aggregates, while the higher electrical conductivity of sludge also contributed to the formation of the DIET process within aggregates [86].

It was shown that the electrical conductivity of biochar was strongly correlated with the amount of aromatic functional groups, and an increase in pyrolysis temperature typically contributed to the formation of conjugated aromatic structures on the surface, which increased the electrical conductivity of biochar [59,87]. Gabhi et al. found that the electrical conductivity of biochar markedly increased with the pyrolysis temperature, from only 15 µS/cm at 600 °C to 370 µS/cm at 800 °C [88]. The electrical conductivity of sludge was continually strengthened with the addition of conductive materials, yet the related benefits to the AD system probably would not increase synchronously with the improvement in conductivity [37]. Studies have shown that the conductivity of conductive pili ranged from only 0 to 18 µS/cm, which suggested that microorganisms for electron transfer probably required less conductivity [37,89]. Hence, it is necessary to investigate the optimum dosage of biochar or preparation technology for AD enhancement in terms of economic feasibility.

#### 4.1.2. Redox-Active Characteristics

Except for electrical conductivity, the redox activity of carbon-based materials was also reported to improve the interspecies electron transfer efficiency of microbes [35]. Particularly, some researchers believe that the rich redox-active functional groups of biochar may play a more important role in the electron transfer between electroactive microbes rather than electrical conductivity [90]. Bu et al. revealed that biochar had greater redox activity at low pyrolysis temperatures, owing to the formation of quinone groups and the adjacent N-doped framework [35]. The quinone groups of biochar resulted in greater redox activity and electron exchange capacity [91,92]. Wang et al. showed that the quinone and the hydro-quinone functional groups on the surface of biochar conferred stronger redox activity, and a linear correlation analysis indicated that the redox-based electron transfer process of microbes correlated strongly with methane production [62]. The high redox activity facilitated biochar potentially acting as an electron transfer mediator, which accelerated the electron transfer between syntrophic microbes [62]. Lu et al. demonstrated that the reduction of carbon dioxide to methane relied primarily on redox cycling based on functional groups of biochar, while biochar with high electron exchange capacity significantly improved methane production by 46.9% [93]. The potential mechanism for the improvement of electron transfer efficiency was the facilitation of the DIET process by unique redox functional groups such as phenazine and quinone [62]. The amount of redox functional groups extremely decreased with the increase in pyrolysis temperature, while the formation of aromatic structures affected the DIET process based on redox-active functional groups [94]. The weakness of redox-active functional groups probably decreased the electron-accepting and electron-donating capacity of biochar during syntrophic metabolism [22].

Cyclic voltammetry (CV) was commonly employed to test the electrochemical characteristics of biochar and revealed the alterations of oxidation/reduction properties on biochar [83]. Meanwhile, the current and integral area of the CV curve illustrated the electron-accepting and donating capacity of redox-active functional groups [95]. The rich redox-active functional groups of biochar provided greater electron exchange capacity and could be a bridge for electron transfer [90,91]. Wang et al. found that biochar at pyrolysis temperatures of 400 °C, containing abundant quinone groups, could boost methane production by 38.1% [59]. Correspondingly, the higher current and integral area of the CV curve (0.50 mA, 0.36 V·mA) indicated the large charging and discharging capacity [59].

Electron change capacity is calculated as the sum of the electron-accepting capacity (EAC) and electron-donating capacity (EDC) [90]. For EAC, Charis et al. found that a carbon-based material was able to accept electrons from the oxidation of organic substances; subsequently, the carbon-based material served as an electron donor for methanogens to facilitate methane production [96]. Some previous studies showed that the improvement of methane production was more dependent on the EDC of biochar than the EAC [62,97]. Qin et al. found a positive linear correlation between the EDC of biochar and the maximum methane production rate [97]. Sun et al. combined the results of correlation analysis and machine learning which also confirmed that the EDC of biochar was positively correlated with cumulative methane production [68]. Wang et al. indicated that biochar with redox activity markedly shortened the lag time and improved the maximum methane production rate [62]. Compared to electrical conductivity, the EDC characteristics conferred from redox-active functional groups probably had a stronger correlation with the DIET process in the syntrophic microbes [62]. Hence, the EDC of biochar was probably the dominant factor to affect methane yield in the AD process.

#### 4.1.3. Electron Transfer System Activity

Electron transfer system (ETS) activity, as an indicator of microbial respiratory activity and sludge bioactivity, could be quantified with the measurement of microbial respiratory chain electron transfer efficiency [94]. Furthermore, the ETS was identified in the whole of respiring anaerobic bacteria, and measurement of activity was commonly considered essential for investigating intracellular electron transfer (IET)-associated metabolic activities such as cellular respiration and dehydrogenase-catalyzed redox reactions [98]. Liu et al. showed that biochar greatly increased the ETS activity by 49.15% in comparison to the control group [98]. Zhao et al. indicated that 1 g/L of biochar significantly improved the ETS activity by 32.6% and a positive correlation between ETS activity and contaminant removal efficiency [94]. Concurrently, the improvement in the ETS activity increased the efficiency of IET, and hence strengthen the exchange of intracellular electrons among redox mediators such as NADH and NAD^+^ [98,99]. The electron transfer system of the cell was classified as IET and extracellular electron transfer (EET) pathways [98]. Li et al. found that the enhancement of IET capacity could accelerate the transfer of more electrons from the oxidation process of electron donors to the EET pathway, thereby improving the efficiency of EET to enhance the biodegradation of contaminants [100]. Lei et al. indicated that biochar increased the ETS activity of sludge from 3.3 to 4.5 μg/mg/h (*p* < 0.05) and improved the biochemical reaction rate and the biotransformation potential of organic micropollutants [60]. Yang et al. showed that biochar significantly increased ETS activity by 35–40% and facilitated the development of a DIET-based methanogenic pathway [1]. ETS activity could represent indirect evidence for the existence of the DIET process in the AD system [101]. Wang et al. suggested that the ETS activity of sludge greatly increased with biochar addition in the AD system, in which the redox-active biochar likely acted as an electron transfer mediator to stimulate the generation of the DIET process, thereby alleviating the inhibition of syntrophic metabolism by high TC pressure [25].

### 4.2. Microbial Mechanism of Biochar-Facilitated DIET

Microbial analysis is important for exploring potential DIET mechanisms. The enhanced abundance of potential DIET microorganisms, facilitated microbial aggregation, and regulated DIET-associated gene expression compose the microbial mechanism of biochar-facilitated DIET (Figure 2).

#### 4.2.1. Enhanced Abundance of Potential DIET Microorganisms

It has been confirmed that biochar can significantly enhance the abundance of potential DIET microorganisms, which affect methane production and contaminant removal in AD systems [37]. For instance, Zhang et al. showed that biochar regulated the microbial community, increased the diversity of archaea, and facilitated the enrichment of *Methanosarcina* [56]. Wang et al. revealed that the relative abundance of Anaerolineaceae was remarkably increased by 14.5% with the addition of biochar, and the activity of metabolic function involved in replication and membrane transport was increased by 24.2–41.9% to facilitate the biodegradation of antibiotics [25]. Moreover, biochar as a stimulator selectively enriched some specific species to enhance the efficient oxidation of VFAs by facilitating the DIET process between syntrophic microbes [25].

The enrichment of electroactive microbes with high electron transfer capability indirectly supported the existence of the DIET process in AD systems [37]. Hitherto, the known syntrophic microbes based on DIET were isolated to a few bacteria such as *Desulfobacula*, *Pseudomonas, Anaerolinea,* and *Geobacter* [37,102,103]. In addition, the well-known syntrophic microbes for methanogenesis were confined to a few archaea such as *Methanosarcina*, *Methanosaeta,* and *Methanothrix* [74]. Li et al. showed that biochar greatly contributed to the enrichment of DIET microbes including *Geobacter*, *Sphaerochaeta*, and *Sporanaerobacter* [2]. These microbes were probably involved in the syntrophic metabolism of organic substances via the DIET process [2]. Jiang et al. indicated that 10 g/L of biochar in dewatered sludge AD of 4% and 1% (v/v) inoculation ratio dramatically increased the relative abundance of *Methanosarcina* by 5.5% and 6.6%, respectively [74]. It was shown that the enrichment of *Methanosarcina* species, defined as DIET microbes, was important for the DIET-based syntrophic metabolism of methane production [104,105]. Yang et al. demonstrated that biochar improved the diversity of bacteria in the AD system, in which biochar increased the relative abundance of *Defluviitoga* and *Thermovirga* by 2.7 and 8.1 times, respectively, while biochar significantly boosted the growth of *Methanothrix*, accounting for approximately 70% of the archaea community [1]. Because *Methanothrix* species were recognized to receive only a few energies in the conversion of acetate to methane [54], Yang et al. believed that biochar could work as an electrical conduit to facilitate the DIET process, which saved the energy consumed to generate the biological electron transfer component (e.g., c-type cytochromes and conductive pili) to promote the growth of *Methanothrix* [1]. The enrichment of microbes that might be involved in the DIET process generally contributed to more efficient methanogenesis in a digester. Li et al. suggested that Pseudomonadaceae, Bacillaceae, and Clostridiaceae were the dominant bacteria in the digester amended with biochar, with a relative abundance of 65.71% that was well above the control of 34.68% [57]. Simultaneously, *Methanosarcina* have enriched on biochar surface and conducted electron transfer with syntrophic bacteria through aromatic functional groups [57].

#### 4.2.2. Facilitated Microbial Aggregation

Extracellular polymeric substances (EPSs) are polymeric substances secreted from microbes and are of significance for microbial adhesion and the generation of biofilm and granular sludge [106]. EPSs are primarily made up of proteins and polysaccharides and bind to humic substances and a few lipids to form polymers [37]. Studies showed that biochar could act as a microbial immobilization carrier, and microbes adhered on the surface generated a unique spatial ecological niche through the secretion of EPSs as a biological skeleton [35,107]. Bu et al. revealed that the total protein and polysaccharide contents of EPSs in AD amended with biochar were greatly increased by 44% and 78.3%, respectively [35]. Biochar stimulated the biosynthesis of cellular EPSs to facilitate the formation of biofilm with a distinctive spatial structure on the surface [108]. Biofilm was categorized into tightly adhered surfaces, loosely existing surfaces, and suspended sludge [108]. Bu et al. found that both suspended and loose microbial colonies on the biochar surfaces were highly enriched with electroactive bacteria, such as *Desulforhabdus* and *Clostridiales*, capable of syntrophic metabolism with methanogens [35]. Moreover, Yan et al. showed that biochar facilitated the selective enrichment of *Trichococcus* and *Methanomicrobiales* on its surface, and it served as a mediator of electron transport, improving the synergy and activity of microbes and further boosting the formation of biofilms [69].

A stable micro-ecosystem provided a favorable environment for electron transfer between microbes. Li et al. showed that the aromatic structure on the biochar surface offered a beneficial environment for the enriched microbes to conduct the DIET process [57]. Wang et al. suggested that the activity of *Methanosaeta* and *Methanosarcina* in AD amended with biochar was only slightly inhibited after a high organic loading shock, and the relative abundance was 2.13 and 3.19 times of control, respectively [31]. The coupling of microbes and biochar created distinctive spatial dispersion, which promoted the aggregation of sludge into structurally stable granules and contributed to alleviating the inhibition of the AD system by high VFAs [31]. DIET microbes could transport electrons through direct contact with conductive pili, in addition to c-type cytochromes and conductive materials for electron transfer [37]. Hence, microbial aggregation contributes to the reinforcement of direct contact with syntrophic microbes, thereby further strengthening the DIET process [37]. Wang et al. found that biochar increased the content of EPSs by 40.5% and excess EPSs facilitated the formation of a robust three-dimensional structure to accelerate the aggregation of sludge into granules [109]. The formation of granular sludge created a situation for the potential DIET process to develop.

#### 4.2.3. Regulated DIET-Associated Gene Expression

DIET-associated genes of microorganisms might notably be upregulated in AD amended with biochar, which was commonly provided as proof to demonstrate the occurrence of DIET among syntrophic methanogenesis [37,58]. Yin et al. found that conductive material facilitated the improvement of diffusible signal factor (DSF) and 3′-5′ cyclic guanosine monophosphate (c-di-GMP)-associated gene abundance in microorganisms, and the signal molecule encoded by these genes probably co-regulated the biosynthesis of type IV pili, which affected the syntrophic metabolism for methanogenesis [41] (Figure 3). Li et al. found that the abundance of gene *pilA*, encoding type IV pili, was significantly improved 3.75 times over 20 days in AD amended with biochar, which suggested that the DIET process between syntrophic microbes was probably conducted through type IV pili [110]. Numerous studies have confirmed that carbon-based materials could stimulate the improvement of *pilA* abundance and promote the generation of biological conductive compounds [111,112]. Cui et al. found that cells with high activity were capable of electron transport via membrane-bound proteins [36]. Biochar as a neutral additive facilitated the enrichment of *Syntrophomonas* and methanogens and triggered the specific expression of relevant enzymes [36]. The abundance of genes for crucial enzymes involved in methanogenic pathways of methylamine, dimethylamine, and trimethylamine increased by 177% in AD amended with biochar [36].

Meanwhile, it was shown that the electrochemical characteristics of biochar enhanced the DIET process among syntrophic microbes and weakened the role of biological conductive compounds [2,37]. Li et al. demonstrated that biochar could serve as an electrical conduit, saving cellular energy for the production of conductive pili or c-type cytochromes, enabling more energy to be transferred toward methanogenesis and biomass synthesis [2]. Park et al. found that the abundance of DIET-associated genes *pilA* and *omcS* declined by 69.4% and 29.4%, respectively, in AD amended with carbon-based material, and methanogenic microbes probably transfer electrons through the electrical conduit during the reduction of carbon dioxide to methane [113]. Furthermore, the metabolic pathways of microorganisms could be used to explore the potential mechanism of DIET and illuminate the role of biochar on the metabolism of electroactive microbes. Qi et al. found that the gene in carbon dioxide reduction was notably elevated by 12.84% in AD amended with biochar and metagenomic analysis of energy metabolism indicating that *Methanosaeta* is positively involved in the reduction of carbon dioxide to methane [58]. According to microbial community analysis, biochar facilitated the enrichment of DIET microbes such as *Pseudomonas*, *Methanosaeta,* and *Methanosarcina* [58]. Studies showed that *Pseudomonas* was able to perform electron transfer with *Methanosaeta* via the DIET process during the reduction of carbon dioxide to methane [114]. Therefore, biochar might act as an electrical conduit for electron transfer between *Pseudomonas* and *Methanosaeta* [58].

## 5. Challenges of Biochar in Practical Application

Biochar possesses many advantages, such as promoting methane production, reducing the lag time of microbial growth, and providing habitat for microorganisms [35,64]. In addition, biochar, with its low price and sustainability, has considerable economic advantages in terms of economic feasibility analysis [64]. Although researchers have widely investigated the effects of different aspects of biochar on the AD system for few decades, most current studies on biochar applied to microbial systems have only focused on laboratory-scale reactors (<5 L) [57,64], while studies on a pilot-scale are still rare. Pilot-scale AD reactors should be considered prior to large-scale engineering applications in order to gain operational experience for practical industrial applications. Shen et al. found that the addition of biochar improved the stability of the microbial community in the pilot-scale continuous-flow AD reactor, which was one of the strengthening mechanisms of AD performance [64], and the addition of 8 g/L biochar improved the volumetric biogas production rate by 43.09% and an increase in methane content of 7.40% [70]. Bona et al. showed that biochar facilitated a rapid increase in the methane concentration of biogas in the initial stage and ultimately improved the methane content by 10% in the pilot-scale reactor [72]. Zhang et al. showed that the average methane production was improved by 37% at the dosage of 15 g/L and the fluctuation in pH was avoided in a 1000 L pilot-scale reactor [75].

Nevertheless, some potential challenges may still remain for practical industrial applications. For instance, the toxicants, such as polycyclic aromatic hydrocarbons (PAHs), persistent free radicals, and heavy metal ions, released by biochar should be considered for large-scale engineering applications [115,116]. Bona et al. indicated that the addition of biochar increased the PAHs content, with a maximum concentration of 8.9 mgPAHs kgTS^−1^, and reduced the bioavailability of biochar-enriched digestate in the field of agriculture [72]. Except for the toxicants, the structural stability of biochar also needs to be considered. The structure of biochar may be damaged after application in microbial systems, resulting in the deterioration of biochar and the reduction of electrochemical activity [117]. Furthermore, although the porosity of biochar can be a habitat for microorganisms, it can adsorb some signaling molecules, such as DSF and c-di-GMP, and inhibit the occurrence of quorum sensing [118]. The effects of biochar with different particle sizes on AD performance have been widely elucidated in laboratory-scale operations. Except for 1–3 cm particle sizes of biochar, Zhang et al. found that biochar with different particle sizes (<100 μm) has no significant difference in methane production [117]. However, the small particle size of biochar density is generally lower than the density of water, resulting in a tendency of biochar to be washed away [70]. Moreover, it is difficult to recycle biochar in practical engineering because of the incorporation of biochar and microbial systems [117]. Lu et al. indicated that biogenic Fe (II), generated from the coupling of Fe-bearing biochar and microorganisms, not only significantly facilitated the reduction of nitrobenzene, but biochar could be separated with magnets from microbial systems [119]. Hence, it is a strategy to solve the difficulty of recycling. In the future, more studies should focus on feasibility tests of process parameters and environmental risks to obtain the optimization of economic and environmental benefits in pilot-scale reactors.

## 6. Conclusions and Prospects

Biochar-facilitated DIET is a promising strategy for enhancing AD performance. However, it is difficult for current technology to clearly illuminate the contribution of biochar-facilitated DIET mechanism for the enhancement of AD. Therefore, we summarized and categorized recent research efforts to support the evaluation of biochar-facilitated DIET for AD enhancement. This review introduced the evolution of DIET and applications of biochar for facilitating DIET to alleviate antibiotic inhibition and enhance methanogenesis. Meanwhile, the electrochemical mechanism of biochar-facilitated DIET showed that AD performance was enhanced by improving the electrical conductivity of sludge, facilitating electron transfer with redox characteristics, and increasing the ETS activity. In addition, the microbial mechanism of biochar-facilitated DIET has also drawn attention. The enhancement of DIET microorganism abundance, microbial aggregation, and expression of DIET-associated genes are all composed of AD reinforcement. Lastly, the challenges of biochar in practical application were discussed. In the future, we hold the opinion that there were several efforts in this field that deserve further exploration:Development of investigation methods. It is necessary to integrate bioinformatic analysis, in situ spectroelectrochemical characterization, and material morphological characterization into a comprehensive analytical tool to further explore DIET strains. Concurrently, the coupling of molecular simulation and density functional theory calculations with conventional microbial analysis is also recommended to explore the potential DIET mechanisms in depth [120].Mechanisms of antibiotic removal. There is a lack of in-depth research on the mechanism of enhancing biodegradation of antibiotics with biochar-facilitated DIET. The construction of biohybrids of biochar and methanogens (e.g., *Methanosarcina barkeri*) explores the mechanism of antibiotics degradation by the DIET process, and the role of antibiotics during electron storage and redistribution at the biotic-abiotic interface [121].Prediction of machine learning models. It is challenging to obtain the optimal operating conditions such as feedstock type, pyrolysis temperature, and biochar dose for biochar-enhanced AD performance due to the existence of a complex bioconversion process. Employing multitask models constructed with machine learning algorithms provides an in-depth comprehension of key factors for biochar to facilitate DIET-enhanced methanogenesis and alleviate antibiotic suppression [122].

## Figures and Tables

**Figure 1 ijerph-20-02296-f001:**
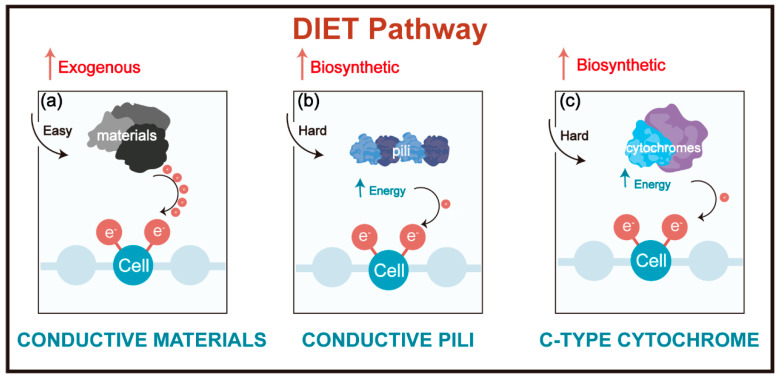
DIET pathways for syntrophic microbes. (**a**) DIET via the conductive material pathway; (**b**) DIET via the conductive pili pathway; (**c**) DIET via the c-type cytochromes pathway.

**Figure 2 ijerph-20-02296-f002:**
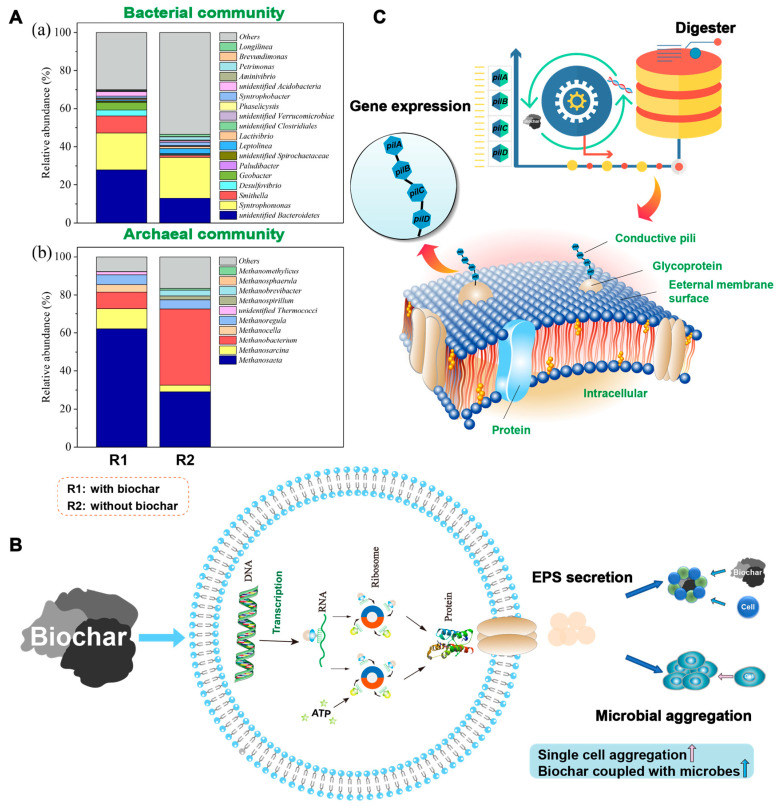
Microbial mechanism of biochar-facilitated DIET. (**A**) Enhanced abundance of DIET microorganisms, modified from Ref. [31], copyright (2021) Elsevier. (**B**) Facilitated microbial aggregation. (**C**) Regulated DIET-associated gene expression.

**Figure 3 ijerph-20-02296-f003:**
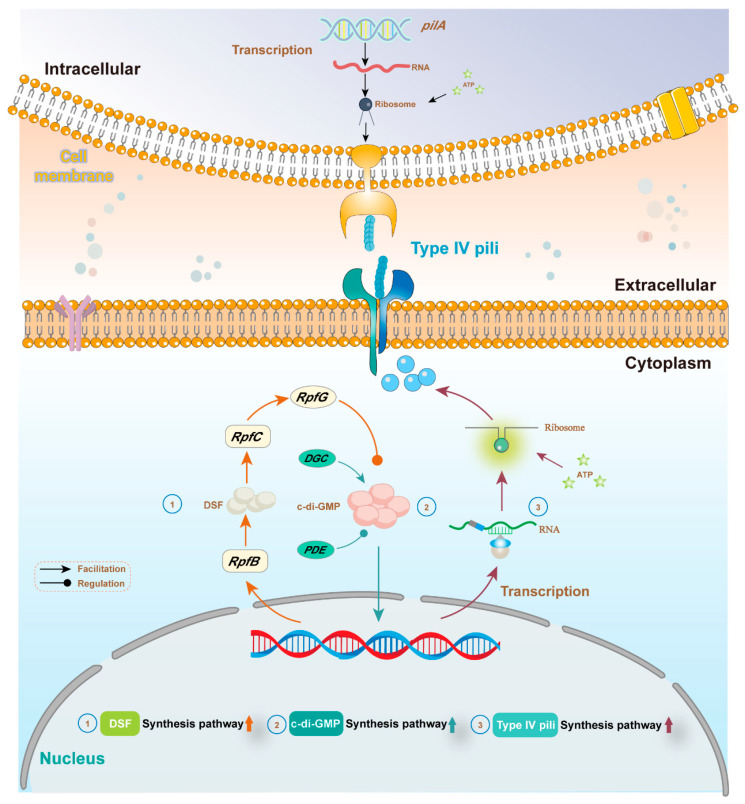
Potential mechanisms of quorum sensing facilitated type IV pili generation, modified from Ref. [41]; copyright (2020) Elsevier.

## Data Availability

Not applicable.

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
