# Peer review of "Biochar Facilitated Direct Interspecies Electron Transfer in Anaerobic Digestion to Alleviate Antibiotics Inhibition and Enhance Methanogenesis: A Review"

_ijerph, 2023, doi:10.3390/ijerph20032296_

Round 1

Reviewer 1 Report

The manuscript “Biochar facilitated direct interspecies electron transfer in anaerobic digestion to alleviate antibiotics inhibition and enhance methanogenesis: A review” by Zhang et al. described the role of interspecies electron transfer (DIET) facilitated by biochar on anaerobic digestion performance and discussed its perspectives.  Overall, the manuscript is interesting and requires revision as follows:

Comments

1.      Line 26, please provide one or two sentences about this review article's significance and critical perspectives points.

2.      Lines 29-31, please add the composition of biogas (enriched in methane) produced in AD.

3.      Lines 34-39, microbial biodiversity and their synchronized association among different steps of the AD process essential for successful operation. Please add a few details about microbes and their properties involved in various AD steps, i.e., Renewable & Sustainable Energy Reviews 150 (2021) 111491.

4.      Lines 42-44, How do antibiotics influence the AD process via microbial community perfectives and mechanisms? Please elaborate in detail, i.e., Bioresource Technology 361 (2022) 127662.

5.      Lines 47-49, Highlight the critical points on the advantages and disadvantages of AD, along with perspectives. In addition, AD-evolved biogas (enriched in methane and carbon dioxide) is considered a major greenhouse gas (GHGs), and its emission have various adverse environmental effects. However, the utilization or mitigation of these GHGs by methanotrophs can be a helpful strategy for generating renewable energy and managing various kinds of biowastes and GHGs, i.e.  Bioresource Technology 367 (2023) 128427; Atmosphere 14 (2023), 120.

6.      Please avoid the use of SD values in the text.

7.      Table 1, please add the biochar properties column also.

8.      Figure 1 and Figure 2, the quality can be improved, i.e., font size and resolution.

9.      Please add one section on a “Pilot scale methanogenesis, and challenges or limitations”. 

Author Response

Response to Reviewer 1 Comments

         Thank you for your kind decision and for the reviewers’ comments concerning our manuscript. Those comments are all valuable and very helpful for revising and improving our paper, as well as the important guiding significance to our researchers. We have studied comments carefully and have made correction which we hope meet with approval. Revised portion are marked in blue in the revised manuscript. The main corrections in the paper and the responds to the editor’s and reviewers’ comments are as following:

Point 1: The manuscript “Biochar facilitated direct interspecies electron transfer in anaerobic digestion to alleviate antibiotics inhibition and enhance methanogenesis: A review” by Zhang et al. described the role of interspecies electron transfer (DIET) facilitated by biochar on anaerobic digestion performance and discussed its perspectives. Overall, the manuscript is interesting and requires revision as follows:

Response 1: Thank you for your kind decision and valuable comments.

Point 2: Line 26, please provide one or two sentences about this review article's significance and critical perspectives points.

Response 2: Thank you for your valuable comment. We have added the suggested content to the revised manuscript as the following:This review elucidated the role of DIET facilitated by biochar in AD system, which would advance our understanding of the DIET mechanism underpinning the interaction of biochar and anaerobic microorganisms. However, the direct evidence for the occurrence of biochar-facilitated DIET still requires further investigation (page 1, lines 25-29).

Point 3: Lines 29-31, please add the composition of biogas (enriched in methane) produced in AD .

Response 3: Thank you very much for your valuable comment. We have added the suggested content to the revised manuscript as the following:The composition of biogas is about 40-70% methane and 30-60% carbon dioxide and small amounts of gaseous water, hydrogen sulfide and ammonia (page 2, lines 35-37).

Point 4: Lines 34-39, microbial biodiversity and their synchronized association among different steps of the AD process essential for successful operation. Please add a few details about microbes and their properties involved in various AD steps, i.e., Renewable & Sustainable Energy Reviews 150 (2021) 111491.

Response 4: We sincerely appreciate the valuable comments. Details about microbes and their properties involved in various AD steps has been added and the reference literature has been added into the revised manuscript as suggested (page 2, lines 40-49).

Point 5: Lines 42-44, How do antibiotics influence the AD process via microbial community perfectives and mechanisms? Please elaborate in detail, i.e., Bioresource Technology 361 (2022) 127662.

Response 5: Thank you for your valuable suggestion. We have benefited a lot from the article that you recommended to us. Microbial inhibition by antibiotics has been suggested as the main reason for decreasing the efficiency of AD system. For example, antibiotics can weaken the ability of microbes to transform complex organic matter into soluble monomers by disrupting cell wall structure, inhibiting protein synthesis, and interfering with cell membrane function. The content and reference literature have been added into the revised manuscript as suggested (page 2, lines 61-67).

Point 6: Lines 47-49, Highlight the critical points on the advantages and disadvantages of AD, along with perspectives. In addition, AD-evolved biogas (enriched in methane and carbon dioxide) is considered a major greenhouse gas (GHGs), and its emission have various adverse environmental effects. However, the utilization or mitigation of these GHGs by methanotrophs can be a helpful strategy for generating renewable energy and managing various kinds of biowastes and GHGs, i.e.  Bioresource Technology 367 (2023) 128427; Atmosphere 14 (2023), 120.

Response 6: Thank you for your valuable advice. We have benefited a lot from the article that you recommended to us. Critical points on the advantages and disadvantages of AD has been added and the reference literature has been added into the revised manuscript as suggested (page 2, lines 48-53).

Point 7: Please avoid the use of SD values in the text.

Response 7: Thank you very much for your valuable comment. We have deleted SD values in the revised manuscript as suggested.

Point 8: Table 1, please add the biochar properties column also.

Response 8: Thank you for your valuable suggestion. We strongly agree that biochar properties are a key factor for its application in AD. Biochar properties such as particle size and BET have been added into Table 1 in the revised manuscript as suggested (pages 7-11).

Point 9: Figure 1 and Figure 2, the quality can be improved, i.e., font size and resolution.

Response 9: Thank you very much for your valuable comment. We strongly agree that high quality graphics can help to improve the readability of an article. According to your suggestion, we have redrawn figures in the revised manuscript (pages 3 and 18).

Point 10: Please add one section on a “Pilot scale methanogenesis, and challenges or limitations”.

Response 10: Thank you for your valuable comment. We have added the section about the challenges of biochar in practical applications based on the comment of “Pilot scale methanogenesis, and challenges or limitations” (pages 19-20, lines 462-503).

Reviewer 2 Report

This paper, entitled Biochar facilitated direct interspecies electron transfer in anaerobic digestion to alleviate antibiotics inhibition and enhance methanogenesis: A review, is a scholarly work and can increase konwledge on this domain. The authors provide an interesting and original review, the content is relevant to IJERPH.

I have some general and specific comments:

- The abstract and keywords are meaningful.

The manuscript is quite well written and well related to existing literature.

- Is the Figure 1 adpated from other figures in other studies or a special creation for this manuscript? There's no reference nor information about his point. This figire is not selfcomprehensible and requires some comments oin the text. Please discuss this figure deeper in the text and explain the different schemes.

- In Table 1, please provide information abot the scale of experiment, is it a labscale or pilot scale? Please complete the Table with such information.

- Please improve readability of Figure 2. The Figure is not easy to read, maybe the different information or schemes could be split into different different figures.

The authors carried out a great work of compilation that is very important and from high interest, theses information coudl generate new knowledge on this domain that is in the current research work. As it, the manuscript is not fully acceptable and requires minor amendments. This paper should be foresure published after these minor revision and amendments. I recommend the following decision: PUBLISH AFTER MINOR REVISION.

Author Response

Response to Reviewer 2 Comments

    Thank you for your kind decision and for the reviewers’ comments concerning our manuscript. Those comments are all valuable and very helpful for revising and improving our paper, as well as the important guiding significance to our researchers. We have studied comments carefully and have made correction which we hope meet with approval. Revised portion are marked in blue in the revised manuscript. The main corrections in the paper and the responds to the editor’s and reviewers’ comments are as following:

Point 1: This paper, entitled Biochar facilitated direct interspecies electron transfer in anaerobic digestion to alleviate antibiotics inhibition and enhance methanogenesis: A review, is a scholarly work and can increase knowledge on this domain. The authors provide an interesting and original review, the content is relevant to IJERPH. I have some general and specific comments: The abstract and keywords are meaningful. The manuscript is quite well written and well related to existing literature.

Response 1: Thank you for your kind decision and valuable comments.

Point 2: Is the Figure 1 adapted from other figures in other studies or a special creation for this manuscript? There's no reference nor information about his point. This figure is not selfcomprehensible and requires some comments in the text. Please discuss this figure deeper in the text and explain the different schemes.

Response 2: Thank you very much for your valuable comment. We strongly agree that high quality graphics can help to improve the readability of an article and apologize for the poor readability of the original figures. According to your suggestion, we have redrawn figures in the revised manuscript (page 3).

Point 3: In Table 1, please provide information about the scale of experiment, is it a labscale or pilot scale? Please complete the Table with such information.

Response 3: Thank you for your valuable suggestion. We strongly agree that the scale of experiment is something that deserves our attention, which has been added into Table 1 in the revised manuscript as suggested (pages 7-11).

Point 4: Please improve readability of Figure 2. The Figure is not easy to read, maybe the different information or schemes could be split into different figures.

Response 4: Thank you very much for your valuable comment. We strongly agree that high quality graphics help improve the readability of an article. According to your suggestion, we have redrawn figures in the revised manuscript (page 18).

Round 2

Reviewer 1 Report

The manuscript quality has been significantly improved after revision.  It can be accepted.